Radiomics combined with clinical features in distinguishing non-calcifying tuberculosis granuloma and lung adenocarcinoma in small pulmonary nodules

Dong Qing 1
Wen Qingqing 2
Li Nan 3
Tong Jinlong 4
Li Zhaofu 5
Bao Xin 6
Xu Jinzhi 1
Li Dandan hmu.cancer.hospital@gmail.com 7
1 Department of Thoracic Surgery at No. 4 Affiliated Hospital, Harbin Medical University , Harbin , China
2 Icahn School of Medicine at Mount Sinai , New York , NY , United States of America
3 Department of Pathology at No. 4 Affiliated Hospital, Harbin Medical University , Harbin , China
4 Department of Medical Imaging at No. 4 Affiliated Hospital, Harbin Medical University , Harbin , China
5 Heilongjiang Institute of Automation , Harbin , China
6 Harbin Medtech Innovative Company , Harbin , China
7 Department of Radiology at Cancer Hospital, Harbin Medical University , Harbin , China
Huisman Henkjan
Electronic publication date: 2022 Oct 19
Publication date: 2022
Volume: 10
Electronic Location ID: e14127
Received 2021 Aug 24; Accepted 2022 Sep 6
Copyright: ©2022 Dong et al.
Copyright year: 2022
Copyright holder: Dong et al.
License: This is an open access article distributed under the terms of the Creative Commons Attribution License, which permits unrestricted use, distribution, reproduction and adaptation in any medium and for any purpose provided that it is properly attributed. For attribution, the original author(s), title, publication source (PeerJ) and either DOI or URL of the article must be cited.
License URL: https://creativecommons.org/licenses/by/4.0/

Keywords: Radiomics, Non-calcified tuberculosis granuloma, Lung adenocarcinoma, Pulmonary nodules, Clinical features

Funding: National Natural Science Foundation of China 61263033 International Science and Technology Cooperation Project of Hainan KJHZ2015-4 Higher School Scientific Research Project of Hainan Province Hnky2015-80 Clinical Study on the Changes of Brain Net Efficiency after Preventive Brain Irradiation in Patients with Limited Stage Small Cell Lung Cancer This work was supported by the National Natural Science Foundation of China (No. 61263033), the International Science and Technology Cooperation Project of Hainan (No. KJHZ2015-4), the Higher School Scientific Research Project of Hainan Province (No. Hnky2015-80), and the Clinical Study on the Changes of Brain Net Efficiency after Preventive Brain Irradiation in Patients with Limited Stage Small Cell Lung Cancer. The funders had no role in study design, data collection and analysis, decision to publish, or preparation of the manuscript.

==============================
Aim

To evaluate the performance of radiomics models with the combination of clinical features in distinguishing non-calcified tuberculosis granuloma (TBG) and lung adenocarcinoma (LAC) in small pulmonary nodules.

Methodology

We conducted a retrospective analysis of 280 patients with pulmonary nodules confirmed by surgical biopsy from January 2017 to December 2020. Samples were divided into LAC group (n = 143) and TBG group (n = 137). We assigned them to a training dataset (n = 196) and a testing dataset (n = 84). Clinical features including gender, age, smoking, CT appearance (size, location, spiculated sign, lobulated shape, vessel convergence, and pleural indentation) were extracted and included in the radiomics models. 3D slicer and FAE software were used to delineate the Region of Interest (ROI) and extract clinical features. The performance of the model was evaluated by the Area Under the Receiver Operating Characteristic (ROC) Curve (AUC).

Results

Based on the model selection, clinical features gender, and age in the LAC group and TBG group showed a significant difference in both datasets (P < 0.05). CT appearance lobulated shape was also significantly different in the LAC group and TBG group (Training dataset, P = 0.034; Testing dataset, P = 0.030). AUC were 0.8344 (95% CI [0.7712–0.8872]) and 0.751 (95% CI [0.6382–0.8531]) in training and testing dataset, respectively.

Conclusion

With the capacity to detect differences between TBG and LAC based on their clinical features, radiomics models with a combined of clinical features may function as the potential non-invasive tool for distinguishing TBG and LAC in small pulmonary nodules.

Introduction

Tuberculosis (TB) is an infectious disease that is caused by a single source (MacNeil et al., 2020). According to statistics, there are about 10 million new TB patients and 1.5 million deaths each year, more than any other infectious disease (Thwaites & Nahid, 2020). Among them, pulmonary TB is the most common, accounting for about 85% of all tuberculosis cases (Reid et al., 2019). Its pathological manifestation is chronic granulomatous inflammation (Yuan & Sampson, 2018). In 2020, 1,930 million new cancer cases and 10 million deaths were estimated worldwide, with approximately 2.2 million (11.4%) new lung cancer cases and 1.8 million (18%) deaths (Sung et al., 2021). LAC is the most common malignant tumor, its prognosis is much worse than tuberculosis, so early diagnosis and treatment are very important. However, it is difficult to distinguish TBG and LAC in chest images, and even nuclear medicine is nonspecific (Fischer, Lassen & Højgaard, 2011; McWilliams et al., 2013). Because both diseases can be shown as solid nodules or masses on imaging studies and have similar radiological features. The confirmative diagnosis of pulmonary nodules is usually biopsy or surgery (Siegel, Miller & Jemal, 2019). However, this invasive examination may lead to possible tissue damage (Pisano et al., 2020). Besides, unnecessary imaging studies may also delay treatment, or miss the best treatment time window (Huo et al., 2019). Therefore, it is expected in clinical practice that a method can be used to monitor pulmonary nodules noninvasively, and may also provide effective support for the diagnosis and treatment of pulmonary nodules. Radiomics is used to extract features from radiological images and make these features in a quantifiable manner. Its purpose is to better or more consistently discover radiological features, and provide objective features that cannot be provided by standard visual image interpretation for quantitative and qualitative density and morphological characteristics of pulmonary nodules (Bi et al., 2019; Peikert, Bartholmai & Maldonado, 2020). Radiomics can be used for auxiliary diagnosis of pulmonary nodules and prognosis prediction of lung cancer (Mu et al., 2020; Hosny et al., 2018). Importantly, radiomics has been applied to evaluate the molecular and clinical features of lung cancer because of its capacity of detecting atypical features in tumor lesions (Grossmann et al., 2017). In this study, we hypothesized that radiomics analysis could distinguish TBG and LAC in small pulmonary nodules based on imaging and clinical features. To test this idea, we extracted the features of small nodules from lung CT using radiomics technology, obtained the radiological model through statistical analysis, and combined it with clinical features. Our goal is to develop a non-invasive method of distinguishing benign and malignant pulmonary nodules using radiomics models in a combination of clinical features.

Materials & Methods

Patients selection

Our research had been approved by the Ethics Review Committee of No.4th Affiliated Hospital of Harbin Medical University (Institutional Review Board that approved number: KY2020-04). Since it was a retrospective study, additional informed consent was waived. Samples that meet all the following criteria were included: (1) Pulmonary tuberculosis or primary LAC confirmed by biopsy or surgical pathology. (2) Enhanced chest CT images that were collected within 1 month before surgery. (3) Isolated non-calcified pulmonary nodules. (4) The maximum diameter was less than 30 mm. Samples were excluded if they did not meet the above criteria. According to the above inclusion and exclusion criteria, we enrolled 280 patients (143 LAC, 137 TBG) who met the inclusion criteria from January 2017 to December 2020. Patients were randomly selected into training and testing data sets by FeAture Explorer (FAE) software based on the TBG or LAC group.

Evaluation of pathology

All specimens were fixed with formalin and stained with hematoxylin and eosin (HE). In order to judge the biopsy results separately, two pathologists with more than 10 years of working experience were blind to the clinical information. All lesions were classified according to the international standard (Rami-Porta et al., 2017). Classification of Pulmonary Adenocarcinoma according to the latest IASLC/ATS/ERS criteria in previous study (Eguchi et al., 2014): (1). Preinvasive lesions (2). Minimally invasive adenocarcinoma (≤3 cm lepidic predominant tumor with ≤5 mm invasion) (3) Invasive adenocarcinoma (4) Variants of invasive adenocarcinoma

CT data collection

Scanning parameters: The second generation gemstone spectral CT (Discovery CT750 HD) of the US General Electric Company was used to perform dual-phase enhanced CT examination of 280 patients. Patients were in the supine position, scan range was from chest entrance to the diaphragm, to ensure full coverage of all lung tissue. A total of 75 mL non-ionic iodine contrast agent Ioversol (350 mgI/ml) was injected with a double-tube high-pressure syringe at a flow rate of 3.5 mL/s. After injection into the elbow vein, the thoracic aorta at the level of tracheal protuberance was automatically selected as the starting point for monitoring. The intelligent tracking technology of the contrast agent was used to determine the starting time of scanning. When the threshold reached 130 Hu, the scanning was automatically triggered. A venous phase scan started at 80 s. Other parameters were as follows: layer thickness was 0.625 mm, frame rotation time was 0.6 s, pitch was 1.375, and tube current was 600 mA.

Image evaluation

The CT appearance including lesion size, location, burr, lobulation, vascular penetration, and pleural involvement was extracted by two radiologists with more than 10 years of imaging diagnosis experience. Other clinical features such as age, gender, and smoking history were obtained from the electronic health records. To keep a subjective clinical judgment, the two radiologists were blind to both baseline information and biopsy results. If there were conflicting opinions, an agreement would be achieved after discussion. For example, an average value of lesion size was taken after discussion if there were conflicting opinions between radiologists.

Tumor segmentation

We loaded CT images into 3D slicer software (version 4.10.0) for manual segmentation (Fig. 1A, left side). The region of interest (ROI) on CT was delineated by a thoracic surgeon with 10 years of lung surgery experience (Fig. 1A, right side). The ROI was then confirmed by another senior radiologist with chest radiograph experience for more than 10 years.

Radiomics feature extraction and model building

We selected 196 cases as the training dataset (96/100 = TBG/LAC) and 84 cases as the testing dataset (41/43 = TBG/LAC). 851 radiomics features were extracted from each ROI and divided into three main categories: (1) First-order features. (2) Shape characteristics. (3) Texture features, including gray level co-occurrence matrix (GLCM) features, grey-level run-length matrix (GLRLM) features, gray level size zone matrix (GLSZM) features, neighborhood grey tone difference matrix (NGTDM) features, and grey level dependence matrix (GLDM) features. Figure 2 showed how Grey Level Histogram worked. FAE applied uniformization automatically to the feature matrix when preprocessing CT data, where each feature vector subtracted its average value and then divided by its length. Since the dimensional feature space was very high, the similarity of each feature pair was compared. If the Pearson Correlation Coefficient (PCC) of one feature pair was greater than 0.99, one of them from the pair was removed. After this preprocessing procedure, the size of the feature space was reduced, and each feature was independent of another. Kruskal Wallis was utilized to explore the important features corresponding to labels. In the FAE software, Pearson and Kruskal Wallis methods were automatically selected in the FAE software and we applied them to the training dataset. To evaluate the relationship between features and labels, we calculated the F value. Afterward, we ranked the top 14 features according to the corresponding F value. These 14 features were chosen by the FAE software based on the highest F value. Eventually, Random Forest Model with the highest AUC value was chosen automatically by FAE software as a classifier from all existing models including Support Vector Machine (SVM), Latent Dirichlet Allocation (LDA), Autoencoder (AE), Random Forest, Logistic Regression-Lasso, Adaboost, Decision Tree, Gaussian Process, Naive Bayes. To determine the hyperparameters of the model (e.g., The number of features), we applied 10 times cross-validation on the training dataset. Therefore, hyperparameters were set according to the model performance on the validation dataset (Fig. 1).

Figure 1 Research method.

Overview of research methods: (A) Collection of chest CT data and ROI delineation. (B) Feature extraction. The image is the gray scale histogram of the lesion. (C) Data analysis. (D) Operation using FAE software.

Figure 2 CT images showed lung adenocarcinoma (LAC) and non-calciûed tuberculous granuloma (TB); (A) and (D) CT scan showed irregular solid nodules (red area) in the left upper lobe; (B) and (E) gray scale histogram of the nodule; (C) LAC with hematoxylin and eosin (H & E) stain, ×400; (F) TB with hematoxylin and eosin (H & E) stain, ×400.

Statistical analysis

We used the Statistical Program for Social Science (SPSS, version 16.0) to test statistical differences in clinical features between LAC and TBG groups. The independence of categorical variables was examined by the Chi-square test and Fisher exact test. To test the continuous variables with normal distribution, a t-test was conducted (P < 0.05 indicates statistical significance). We used the Chi-square test for categorical variables such as location, smoking, and other clinical features. The performance of the model and quantitative analysis were evaluated by the ROC curve and AUC (Fig. 1C), respectively. Sensitivity, specificity, positive predictive value (PPV), and negative predictive value (NPV) were calculated when the Youden index was maximized to its cut-point value. We estimated 95% confidence intervals for 1000 samples by bootstrapping. All the above processes were operated via FeAture Explorer Pro (FAEPro, V0.3.5, Fig. 1D) on Python (3.7.6) according to the software operation reference related literature (Song et al., 2020).

Results

Clinical features

Table 1 listed the statistical test results in the training dataset and testing dataset. There were 196 patients in the training dataset, including 96 males (age range: 40–79 years old, average age: 64.53 ± 9.21 years old) and 100 females (age range: 33–72 years old, mean age: 56.06 ± 10.98 years). The testing dataset included 84 patients with 42 males (age range: 41–79 years old, average age: 63.71 ± 10.22 years old) and 42 females (age range: 33–73 years, mean age: 58.65 ± 10.71 years). Patients’ gender and age were significantly different in the LAC group and TBG group in both datasets (Training dataset, Gender: P = 0.001, Age: P = 0.006; Testing dataset, Gender: P = 0.016, Age: P = 0.005). However, TB and LAC were indistinguishable by some clinical features such as smoking status. For example, there was no statistical difference between smoking history and patients’ LAC or TB status (Training dataset, P = 0.15; Testing dataset, P = 0.536). In CT appearance, the lobulated shape was found to show a significant difference in the LAC group and TBG group in the training dataset (P = 0.03) and the testing dataset (P = 0.030). The rest CT features did not show any statistical difference in two groups, including size (Training dataset, P = 0.60; Testing dataset, P = 0.67), location (Training dataset, P = 0.910; Testing dataset, P = 0.43), spiculated sign (Training dataset, P = 0.97; Testing dataset, P = 0.79), vessel convergence (Training dataset, P = 0.40; Testing dataset, P = 0.43), and pleural indentation (Training dataset, P = 0.34; Testing dataset, P = 0.85). These CT features were not distinguishable between LAC and TB in the model.

Table 1 Clinical characteristics and CT findings in LAC and TB.

Characteristic	Training data set (n = 196)	P	Test data set (n = 84)	P	
	LAC(100)	TB(96)		LAC(43)	TB(41)		
Gender			*0.001			*0.016	
Male	37	59		16	26		
Female	63	37		27	15		
Age (mean ± SD, years)	64.53 ± 9.21	56.06 ± 10.98	*0.006	63.71 ± 10.22	58.65 ± 10.71	*0.005	
Smoking history			0.148			0.536	
Absence	69	75		29	25		
Presence	31	21		14	16		
Size (mean ± SD, mm)	19.81 ± 7.47	18.69 ± 5.44	0.595	20.71 ± 7.62	19.03 ± 9.01	0.667	
Location			0.910			0.425	
Upper and middle	68	66		28	30		
Lower	32	30		15	11		
Spiculated sign			0.967			0.791	
Absence	57	55		25	25		
Presence	43	41		18	16		
Lobulated shape			*0.034			*0.030	
Absence	36	49		14	23		
Presence	64	47		29	18		
Vessel convergence			0.400			0.425	
Absence	44	48		22	21		
Presence	56	48		21	20		
Pleural indentation			0.337			0.884	
Absence	30	35		13	13		
Presence	70	61		30	28		
Notes.

The differences were assessed with the Wilcoxon rank sum test or Pearson chi-squared test.

CT computed tomography

LAC lung adenocarcinoma

TB pulmonary tuberculosis

SD standard deviation

* P < 0.05.

Feature selection and radiological model construction

Table 2 illustrated the prediction performance of the training dataset and testing dataset. The accuracy of the training data set was 0.781, AUC was 0.834 (95% Confidence Interval = 0.7712–0.887), NPV was 0.782, PPV was 0.779, sensitivity was 0.771, and specificity was 0.790. Accuracy of the testing dataset was 0.726, AUC was 0.751 (95% confidence interval = 0.6382–0.853), NPV was 0.794, PPV was 0.680, sensitivity was 0.829, and specificity was 0.628. Table 3 showed features with the 14 highest AUC values on the testing dataset (Table 3 and Fig. 3). In addition, the ROC curve was shown in Fig. 4 (Training dataset AUC = 0.834; Testing dataset AUC = 0.751).

Table 2 Clinical statistics in the diagnosis.

	Accuracy	AUC	AUC 95% CIs	NPV	PPV	Sensitivity	Specificity	
Training data set	0.7806	0.8344	0.7712–0.8872	0.7822	0.7789	0.7708	0.7900	
Test data set	0.7262	0.751	0.6382–0.8531	0.7941	0.68	0.8293	0.6279	

Table 3 The rank of selected features.

Features	Rank	
original_firstorder_90Percentile	1	
original_firstorder_Energy	2	
original_firstorder_Mean	3	
wavelet-HHL_firstorder_Median	4	
wavelet-HHL_glcm_ClusterProminence	5	
wavelet-HHL_glcm_Imc1	6	
wavelet-HHL_glcm_Imc2	7	
wavelet-HHL_gldm_DependenceEntropy	8	
wavelet-HHL_glrlm_RunEntropy	9	
wavelet-HHL_glszm_GrayLevelNonUniformityNormalized	10	
wavelet-HHL_glszm_SizeZoneNonUniformityNormalized	11	
wavelet-HHL_ngtdm_Busyness	12	
wavelet-HHL_ngtdm_Strength	13	
wavelet-LLH_glcm_MCC	14	

Figure 3 Features selection.

Fourteen features selection (above the yellow line).

Figure 4 ROC curve selection.

ROC curve of the model.

Discussion

The article discussed a non-invasive diagnostic method for distinguishing non-calcifying tuberculosis granuloma from lung adenocarcinoma. The results of this study showed that age, gender, and lobulation were important predictors for distinguishing the LAC group and TB group  (Cui et al., 2020). On the one hand, the average age of patients in the TB group was lower than that in the LAC group, which may be explained by the fact that LAC is a malignant tumor, which is common in elderly patients. On the other hand, the number of female patients in the LAC group was more than that in the TB group, whereas the number of male patients in the LAC group was more than that in the TB group. The gender imbalance in the two groups may lead to statistical differences. It could be explained by the fact that females are prone to LAC compared to males, and males are more susceptible to TB compared to females (Marçôa et al., 2018). Radiomics is a process that transforms the subjective evaluation of images into objective quantitative data. Many studies have shown that it can be used as a non-invasive method to predict the benign and malignant effects of pulmonary nodules (Feng et al., 2020b; Xu et al., 2019; Wilson & Devaraj, 2017). These objective data cannot be identified visually but can be determined in a computer-aided manner. The CT appearance ‘lobulated shape’ in this study was statistically different in both groups. This feature can reflect the heterogeneity within pulmonary nodules and help to identify benign and malignant nodules (Jiang et al., 2021). In this study, 196 cases were selected as the training dataset and 84 cases were chosen as the testing dataset. A total of 851 radiomics features were extracted from each ROI randomly and automatically by software, including 18 first-order features, 14 shape features, 24 gray level co-occurrence matrices, 16 gray area size matrices, 16 gray level travel matrices, five domain gray difference matrices, 14 gray level correlation matrices, and 744 wavelet features. The features were sorted according to the corresponding F value, and the first 14 features are selected according to the verification performance. There are three firstorder features, including original _ firstorder _ 90Percentile, original _ firstorder _ Energy and original _ firstorder _ Mean. The first-order features stand for the difference in the distribution of individual prime parameter values, which reflects the difference in the density of lesions. This is the density difference in internal space between lung adenocarcinoma and non-calcified granuloma, which is difficult to identify from the eyes since it is a high-dimensional spatial feature. These features are related to gray matrix parameters. This indicates that the change of gray level in CT images of lung lesions may potentially contribute to the differential diagnosis of lung adenocarcinoma and non-calcified granuloma (Cui et al., 2020). Random forest was used as a classifier in the model because of its highest AUC value among all models. Lung cancer and granuloma were commonly found in the upper lobe in this study. This may be due to changes in lobulation caused by lung cancer infiltration. However, chronic inflammation may also have similar characteristics. This could explain the reason for the relatively low AUC in the results. The AUC of the training dataset and the testing dataset were 0.834 and 0.751, respectively. The AUC of the training dataset is 0.834 compared to 0.751 in the AUC of the testing dataset. The NPV, PPV, sensitivity, and specificity have high similarities when compared to previous studies of its kind (Feng et al., 2020a; Chen et al., 2020). It may not be appropriate to observe lung cancer for a long time without providing treatment, but suspected nodules that grow slowly are not easily identifiable with imaging studies without a sufficient waiting period. In addition, lung cancer and granuloma cannot be accurately distinguished in PET scans as well (Du et al., 2021). Although the gold standard for lung cancer diagnosis is the surgical biopsy, it is considered overtreatment if the nodule is a granuloma. On the contrary, conservative treatment may delay the timely treatment for lung cancer. Overall, it is difficult to distinguish benign and malignant pulmonary nodules merely using a lung CT scan. Physicians have been seeking a non-invasive examination to solve this problem. Radiomics, in combination with clinical features, shows its potential to be used as an effective tool to assist radiologists to distinguish benign and malignant pulmonary nodules. However, we have several limitations in this study. Firstly, it was a retrospective analysis. The sample size was relatively small and selection bias could be a potential issue. More high-quality samples are needed to prove the validity of the study in the future. Secondly, selected patients who had surgeries were more likely to be patients diagnosed with malignant tumors. Future research should maintain a relatively equal number of pathology results in both LAC and TB groups. Thirdly, different CT scans may affect the quality of image parameters. Therefore, thin-layer CT scanning (with a value of 0.625 mm) was adopted, and Radiomics normalization preprocessing was used to improve the quality of the data.

Conclusions

In summary, radiomics combined with clinical features is a possible non-invasive tool to distinguish non-calcifying tuberculosis granuloma and lung adenocarcinoma in small pulmonary nodules. The application of this combination has a great potential to decrease overdiagnosis and overtreatment in the future.

Supplemental Information

Data S1 Retrospective analysis of lesion sites

280 lesion sites were analyzed and data was obtained through feature extraction

Click here for additional data file.

Additional Information and Declarations

Competing Interests

Author Contributions

Human Ethics

Data Availability

Xin Bao is employed by Harbin Medtech Innovative Company.

Qing Dong conceived and designed the experiments, authored or reviewed drafts of the article, and approved the final draft.

Qingqing Wen performed the experiments, authored or reviewed drafts of the article, and approved the final draft.

Nan Li conceived and designed the experiments, prepared figures and/or tables, authored or reviewed drafts of the article, and approved the final draft.

Jinlong Tong performed the experiments, prepared figures and/or tables, and approved the final draft.

Zhaofu Li performed the experiments, authored or reviewed drafts of the article, and approved the final draft.

Xin Bao analyzed the data, prepared figures and/or tables, and approved the final draft.

Jinzhi Xu analyzed the data, authored or reviewed drafts of the article, and approved the final draft.

Dandan Li conceived and designed the experiments, prepared figures and/or tables, and approved the final draft.

The following information was supplied relating to ethical approvals (i.e., approving body and any reference numbers):

No. 4th Affiliated Hospital of Harbin Medical University granted Ethical approval to carry out the study within its facilities (KY2020-04).

The following information was supplied regarding data availability:

The raw data is available as Supplemental File.

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
