# Peer review of "Radiomics combined with clinical features in distinguishing non-calcifying tuberculosis granuloma and lung adenocarcinoma in small pulmonary nodules"

_PeerJ, doi:10.7717/peerj.14127_

## Round 0.1 · original submission · Major Revisions

I agree with the reviewers that substantial changes need to be made to this manuscript.

1. Add clear intended use in the introduction. How is your AI to be used?
2. Add a hypothesis in the introduction. What is it that your experiment should prove?
3. Rephrase your conclusion as a test performance AUC of 0.75 does not mean that you developed an effective non-invasive tool.
4. Fix the very many spelling and grammar issues.

·

Basic reporting

Grammatical errors where words miss letters or sentences are structured in an odd manner can be found throughout the text. E.g:

Methods
Line 72: acording should be according.

Discussion
line 208: ove all should be overall

Additional points:
Introduction:
Line 39: Please change the line about radiomics being a new method, due to its introduction in 2014 it can’t really be called new anymore.
Line 39/40: Consider rewriting the sentence: “It refers to the identification, extraction, quantification, and analysis of image features from radiological images.” Technically radiomics only extracts features in an effort to quantify “invisible” information. For identification and analysis other tools are needed.

Experimental design

Abstract:
Methods line 6: Please correct the line about the division, currently it is divided into two training datasets. This should probably be one training and one test dataset.

Methods:
Patient selection from line 52: Please clean up the selection and exclusion criteria since some of them are redundant. For instance if samples meet: “( 1 ) Pulmonary tuberculosis or primary LAC confirmed by biopsy or surgical pathology.” It is not needed to state the exclusion: “( 1 ) pathological specimen were not able to support the diagnosis of tuberculosis, primary LAC, or coexistence of tuberculosis and primary LAC.” (2)&(2) can be put together while (3)&(3) and (4)&(6) are redundant.

Line 72: Please provide some more information on the pathology grading, do the samples get some kind of score? Is this score then used as a label or are binary classes created?

Line 87: Please clarify the decision process on continues CT features like the lesion size, how is a consensus reached here? Why not take the average of both? If this is done please change the text accordingly.

Line 100: Perhaps better to change “radiological features” to “radiomics features”. Radiological is rather broad and seems to suggest something different than radiomics.

Line 106: Please explain the use of SMOTE, the distribution of TBG/LAC in both the training and test dataset ( 96 / 100 = TBG / LAC , 41 / 43 = TBG / LAC ) is very balanced already. SMOTE is used for severe label imbalances e.g. 100+/10, its use here seems unwanted.

Line 107: Please explain why the feature vector was normalized. Since one CT scanner was used and Hounsfield units are rather meaningful I do not see the need for it. And even then why not normalize image based before the extraction of the radiomics features?

Line 109-123: Current setup for feature selection and model creation is confusing and seems very sensitive to bias. Issues:
- It is unclear whether Pearson and Kruskal Wallis were done on the full dataset or just the training dataset.
- It seems like the pearson and kw steps are done totally separate from the cross validation while the recommended approach would be to tune the whole pipeline. Please explain the choice for separate pearson and KW, one could currently argue that the pearson step(and KW in a lesser amount) removes features that might have a big effect on end performance. Which would be visible when tuning the total pipeline.
- Why not pick a multivariate or ensemble based feature selection approach? Does both the current steps (but better) and is easier to create a pipeline and avoid bias.
- Please clarify the “verification performance” used for KW
- Why were 14 features selected, and how is it possible that the number of features is again tuned for the model selection? The number of features should technically be a hyperparameter of the feature selection, while true hyperparameters are things like trees, max depth (random forest), C, gamma (SVM) etc.
- The different model options are dropped on the reader without any clarification whatsoever, I can deduce that this is linked to FAE but please provide more clarification about the options and use of FAE.

Statistical analysis line 124-130: It is unclear to the reader why all of these tests are done in SPSS on the clinical features and what the goal of said tests is. Please clarify.

Statistical analysis line 130: Accuracy can be removed from the performance analysis, it is redundant when including AUC and the other metrics.

Validity of the findings

Results:
Clinical characteristics and CT features line 137-154: Again unclear what this information adds to the goal of distinguishing TBG from LAC. Some of the characteristics have significant differences between both groups which is a start but then it seems like nothing is done with these features? Additionally the test set is also used to test significance? One would expect to select interesting clinical characteristics on the training dataset and then combine them with radiomics features in a model. This selection of interesting clinical features should be performed in a feature selection + model pipeline together with the radiomics features while tuning the total pipeline.

Line 155 – 163: please clarify this entire prargraph, currently it is unclear where these scores originate. Which model? Are clinical features added?

Please stick to one amount of decimals after the comma, 2 is more than enough for the current purpose.

Discussion:
Line 165: “It is a combination of radiomics and clinical characteristics. This may help doctors solve the problem of distinguishing two common pulmonary nodules” Both statements are unclear or not proven in the current paper, please clarify or remove.

Line 180: “Radioactive characteristic” please rewrite radioactive has a different meaning.

Line 192: There is nothing in the methods about wavelet features. Please add the settings and its use to the relevant part of the text.

Line 194: First mention of the use of Random forest as the final model, please add more information about the selection to the results.

Conclusion:
The total claim made in the conclusion is unfortunately unproven in the current setup of the paper.

Additional comments

Summary:
While the clinical issue and idea are interesting, major methodical issues and lacking information about important parts of the process make all of the claims in this paper unproven. Points regarding methods, results and discussion are especially important.

·

Basic reporting

Please see my additional comments.

Experimental design

Please see my additional comments.

Validity of the findings

Please see my additional comments.

Additional comments

The manuscript entitled “Radiomics combined with clinical features in distinguishing non-calcifying tuberculosis granuloma and lung adenocarcinoma in small pulmonary nodules” by Qing Dong et al. established a diagnostic standard using radiomics and clinical features to differentiate tuberculosis granuloma (TBG) and lung adenocarcinoma (LAC) in small pulmonary nodules. This study is of potential clinical significance and has major issues as below:
1. In Abstract, “We assigned them to a training dataset (n=196) and a training dataset (n=84)”. Should it be "testing dataset"?
2. In Abstract, Why did authors say “Clinical characteristics (eg. gender, age, smoking, etc.) and CT features including size, location, spiculated sign, lobulated shape, vessel convergence, and pleural indentation were included in the model”, while only age, gender and lobulated shape were significantly different between LAC and TB groups?
3. Figure panels were not properly referred in the manuscript. For example, it should be “We loaded CT images into 3D slicer software (version 4.10.0) for manual segmentation (Figure 1A). The region of interest (ROI) on CT was delineated by a thoracic surgeon with 10 years of lung surgery experience (Figure 1B).”
4. “We selected 196 cases as the training dataset (96 / 100 = TBG / LAC) and 84 cases as the testing dataset (41 / 43 = TBG / LAC)”. How were cases selected as the training set or testing set? What was the standard?
5. In text line 100, which are the 851 radiological features? Since Radiomic features are critical in this manuscript, authors should explain in details how these features are selected.
6. Figure 1D is the major data of this manuscript. It should be thoroughly described and interpreted in the Results session, rather than Methods.
7. Figure 2 was not referred and described in the manuscript.
8. Were Table 1 and 2 mistakenly switched?
9. In text line 163, where is Figure 4?
10. In text 208 and 209, it should be “Overall”.
11. In text line 217, “patients who had surgeries were more inclined to those with malignant tumors”. What does it mean?
12. What is the read-out of the diagnostic model? Is it a cut-off value calculated from the training set?

---

## Round 0.2 · Major Revisions

Please respond to the editorial comments in your rebuttal. Miss intended use of AI, miss the hypothesis. It is now a description of running some radiomics software. To be a scientific paper it needs a valid experiment based on a hypothesis. Like is it better than clinicians? Is it better than existing AI? Why? Is the performance enough to change clinical practice? Etc, etc...

---

## Round 0.3 · Minor Revisions

Please respond and address the remaining issues that the reviewers identified. Furthermore, please address my comments by not regarding them as answers in a rebuttal, but actually make changes in the manuscript. For example at the end of the introduction you write:

"In this study, we distinguished TBG and LAC in small pulmonary nodules based on imaging omics and clinical features."

You cannot do that. What you should write is:

In this study, we hypothesize that radiomics analysis can distinguish TBG and LAC in small pulmonary nodules based on imaging and clinical features.

You claim that radiomics is better. Than what? Radiologists? You did not do such study.

Furthermore, you state "Our purpose is to overcome the shortcomings ". What shortcomings?

You conclude that your AI can be non-invasive tool to diagnose these tumors. Do you mean it should run stand-alone? Or assisted reading?

·

Basic reporting

Thank you for the changes to my previous comments, total paper has improved by quite a margin. To finalize please change the following grammar issues:

Paragraph title: "Imagine evaluation" should probably be "Image evaluation"

“Radiomics feature extraction” paragraph:

( 2 ) Shape charateristics. Should be “characteristics”

Uniformlization should be “Uniformization”

Experimental design

No further comments

Validity of the findings

No further comments

Additional comments

No further comments

·

Basic reporting

Please refer to the text lines where the revision has been added. It is confusing only to say "This has been addressed and details have been added to the article" or "Details were added in the article in the discussion".

I understand authors have inclusion and exclusion criteria to select patients. However, What method was used to assign patients into the training or testing sets? Was it random?

Experimental design

NA

Validity of the findings

NA

Additional comments

NA

---

## Round 0.4 · accepted · Accept

All comments addressed and ready for publication